# Risk Factors for the Development of Post-Infectious Bronchiolitis Obliterans in Children: A Systematic Review and Meta-Analysis

**DOI:** 10.3390/pathogens11111268

**Published:** 2022-10-31

**Authors:** Eun Lee, Suyeon Park, Kyunghoon Kim, Hyeon-Jong Yang

**Affiliations:** 1Department of Pediatrics, Chonnam National University Hospital, Chonnam National University Medical School, Gwangju 61469, Korea; 2Department of Applied Statistics, Chung-Ang University, Seoul 06974, Korea; 3Department of Biostatistics, Soonchunhyang University College of Medicine, Seoul 04401, Korea; 4Department of Pediatrics, Seoul National University Bundang Hospital, Seongnam 13620, Korea; 5Department of Pediatrics, Seoul National University College of Medicine, Seoul 03080, Korea; 6Department of Pediatrics, Soonchunhyang University Seoul Hospital, Soonchunhyang University College of Medicine, Seoul 04401, Korea

**Keywords:** post-infectious bronchiolitis obliterans, children, risk factors

## Abstract

Post-infectious bronchiolitis obliterans (PIBO), one of the major complications of respiratory tract infection, is commonly underdiagnosed. To identify the risk groups that may develop PIBO and avoid misdiagnoses, we investigated the risk factors associated with the development of PIBO. We searched PubMed, Embase, and MEDLINE databases for studies that included risk factors for the development of PIBO published from inception to 13 June 2022. We limited our search to studies that reported the estimates of odds ratio (OR), hazard ratio (HR), or relative risks for developing PIBO. A fixed-effect and a random-effect model were used. We included seven studies reporting data on the risk factors for PIBO in 344 children with PIBO and 1310 control children. Twenty-two variables, including sex, age, respiratory pathogens, symptoms, laboratory and radiologic findings, and mechanical ventilation, were mentioned in at least one study. The significant risk factors mentioned in two or more studies included elevated lactate dehydrogenase levels, pleural effusion, hypoxemia, sex, and mechanical ventilation. The significance of the duration of hospitalization and fever as risk factors for PIBO differed when the studies were classified according to the statistical method. In addition, the risk factors differed according to respiratory infection pathogens. This meta-analysis identified potential risk factors associated with the development of PIBO. The results of this study highlight the importance of avoiding misdiagnosis and help establish management strategies for patients at a high risk of developing PIBO.

## 1. Introduction

Post-infectious bronchiolitis obliterans (PIBO) is a chronic and irreversible obstructive airway disease that results from the insult on the small airways following a lower respiratory tract infection [1]. The clinical spectrum of PIBO is diverse, ranging from asymptomatic with fixed obstruction on spirometry to severe respiratory distress requiring continuous oxygen supplementation. Pathologically, PIBO is characterized by peribronchiolar fibrosis with diverse degrees of constructions in the lumen of small airways [2]. Diverse respiratory pathogens, including adenovirus and *Mycoplasma pneumoniae* (MP), can cause the development of PIBO [2].

Although there is a lack of effective treatments for PIBO, a delay in the diagnosis of PIBO reduces the therapeutic effect of potential management strategies since the management strategies have limitations in reversing the progression of peribronchiolar fibrosis [3]. Therefore, the early recognition and diagnosis of PIBO are essential in improving clinical outcomes and reducing the disease burden. However, the time interval between respiratory infections and the onset of symptoms, the limitations in the performance of a pulmonary function test in infants and younger children, and the non-specific symptoms in some PIBO cases delay the diagnosis of PIBO.

The pathophysiologies of respiratory tract infections, including adenovirus and MP, are different according to pathogens [4], and therefore, factors related to the development of PIBO might differ depending on the pathogens. Therefore, identifying risk factors for the development of PIBO, especially according to the pathogens, is necessary to improve the clinical outcomes of PIBO. However, there have been no meta-analysis studies on the risk factors of PIBO with consideration of the pathogens. Therefore, this study involving a systematic review and meta-analysis aims to identify the risk factors associated with the development of PIBO and discuss the differences in the risk factors of PIBO according to the respiratory pathogens.

## 2. Materials and Methods

### 2.1. Literature Search Strategy

This systematic review and meta-analysis were conducted in accordance with the Preferred Reporting Items for Systematic Review and Meta-analyses (PRISMA) reporting guidelines [5]. We searched PubMed, Cochrane Library, and Embase databases for relevant studies from inception until 2 June 2022 by using the search term “bronchiolitis obliterans”. This meta-analysis study was not registered.

### 2.2. Study Selection

Studies satisfying all of the following criteria were included in this meta-analysis: (1) studies that included children diagnosed with PIBO according to the diagnostic criteria; (2) studies that included a control group, defined as the presence of respiratory tract infections without PIBO and patients with PIBO; (3) studies that showed complete risk estimate data and clear outcomes; (4) studies that reported estimates of odds ratio (OR), hazard ratio (HR), or relative risk (RR) with corresponding 95% confidence intervals (CIs); and (5) original articles with cross-sectional, case–control, or randomized controlled trials. Studies were excluded from the present meta-analysis if: (1) the study was a comment, case report, abstract, editorial, letter, or review; (2) there was no description of OR, HR, or RR with 95% CIs; (3) there was no accessible full text; (4) the literature was not published in English; and (5) the study had an uncertain definition for the diagnosis of PIBO.

### 2.3. Data Extraction

Data were collected on study author, publication year, study period, study design, study country, participants’ age, participants’ sex, diagnostic criteria for PIBO, respiratory pathogens, risk factors, and risk estimates, including RR, HR, or OR with 95% CIs. If data were missing, the corresponding authors were contacted. Two reviewers (E.L. and H.J.Y) independently screened articles for eligibility, and disagreements about whether specific articles should be included in our analyses were resolved by discussion based on a consensus.

### 2.4. Quality Assessment

Two reviewers (E.L. and H.J.Y) independently assessed the included studies for risk of bias in a sample population, sample size, participation rate, outcome assessment, and analytical methods to control for bias using the Strengthening the Reporting of Observational Studies in Epidemiology (STROBE) reporting guidelines in five items [6,7]. Disagreements were resolved by discussions between the two authors (E.L., and H.J.Y).

### 2.5. Publication Bias

Publication bias is generally recommended when ten or more studies are included and can be evaluated from three or more studies [8]. In the present meta-analysis, publication bias could not be applied because most meta-analyses included two studies.

### 2.6. Data Analysis

The estimates (OR or HR) with a 95% confidence interval (CI) in several papers were calculated for potential risk factors for the development of PIBO. We extracted the estimates and 95% CIs from the multivariate analysis. If the study provided the univariate analyses without information on the multivariate analyses, the univariate OR or HR was obtained. The extracted estimates and 95% CI were converted into beta and standard error (SE) for analysis and the meta-analysis was conducted for variables reported in at least two studies. We used a random-effect model to calculate the ORs or HRs. An *I*^2^ statistic was used to assess heterogeneity in the results of individual studies, and an *I*^2^ > 50% was used as the threshold indicating significant heterogeneity. A *p* value < 0.05 was considered statistically significant. Forest plots were constructed using summary statistics for risk factors that included two or more studies. The statistical analyses were conducted using R version 3.4.1 and Rex (Version 3.6.0, RexSoft Inc., Seoul, Korea).

## 3. Results

### 3.1. Study Selection and Characteristics

Our initial literature search identified 13,358 studies (Figure 1). After applying the eligible criteria, seven articles were included in the quantitative analysis. The seven case–control studies included 344 children with PIBO and 1310 control children. PIBO was diagnosed based on the HRCT and compatible symptoms [1,2]. Two studies reported risk factors of PIBO, which developed after MP infections [9,10] and four studies reported risk factors of PIBO, which developed after adenovirus infections [11,12,13,14]. The remaining study reported risk factors of PIBO, which developed after diverse respiratory pathogens [15]. Three studies investigated the risk factors of PIBO, which developed after pneumonia [9,13,14] and two investigated the risk factors of PIBO, which developed after bronchiolitis [10,15]. One study included patients with acute lower respiratory tract infections [11] and the remaining study included patients with acute respiratory infections [12].

Two studies used logistic proportional hazard models [12,14], and the other five used logistic regression analysis. Twenty-two risk factors were mentioned at least once, and nine risk factors were mentioned in more than two studies (Table 1). Three risk factors, including duration of fever and hospitalization and mechanical ventilation, were analyzed using the logistic proportional hazard model and logistic regression analysis, resulting in different results for the duration of fever and hospitalizations.

### 3.2. Study Quality

All included studies were assessed for methodological quality using the Newcastle-Ottawa Scale (NOS) (Table 2). The NOS score for the included studies was nine, indicating that all the studies were of high quality.

### 3.3. Risk Factors for the Development of PIBO

The meta-analysis was performed on three risk factors with HRs with 95% CIs (Figure 2A) and nine risk factors with ORs with 95% CIs (Figure 2B).

Three studies indicated that males were at a higher risk of developing PIBO (OR, 1.783; 95% CI, 1.196–2.653). One of the studies reported risk factors of PIBO developing after MP pneumonia [9], and one of the other studies reported risk factors of PIBO developed after adenovirus-induced acute lower respiratory tract infections [11]. The remaining study showed risk factors of PIBO developed after bronchiolitis caused by various respiratory pathogens [15].

Two studies reported the relation between LDH levels and the development of PIBO after MP infections, which showed that higher LDH levels, measured at the time of MP infection, were associated with the development of PIBO (OR, 1.001; 95% CI, 1.000–1.002). There was no significant heterogeneity (*I*^2^ = 51.9%, *p* = 0.124).

A pleural effusion during MP infection was associated with the development of PIBO (OR, 2.851; 95% CI, 1.266–6.421). The heterogeneity was not significant (*I*^2^ = 51.9%, *p* = 0.149).

Two studies reported the association between hypoxemia and PIBO development (OR, 14.239; 95% CI, 4.231–47.918) [10,13]. One study reported a significant association between hypoxemia and the development of PIBO in MP bronchiolitis using multivariate logistic regression analysis [10], whereas the other study showed a significant association between hypoxemia and the development of PIBO in adenovirus pneumonia [13]. Heterogeneity was considered insignificant (*I*^2^ = 0.0%, *p* = 0.149).

Four studies reported the association between mechanical ventilation and PIBO development [11,12,14,15]. Three studies investigated the associations of PIBO and mechanical ventilation in adenovirus infections [11,12,14] and the remaining study elucidated the association of PIBO and mechanical ventilation in acute bronchiolitis caused by diverse respiratory pathogens [15]. The results from the two studies using a logistic proportional hazard model showed that mechanical ventilation in adenovirus infections was associated with the development of PIBO (HR, 3.314; 95% CI, 1.274–8.626). In addition, data extracted from the multivariate logistic regression analysis showed a significant association (OR, 3.377; 95% CI, 2.185–5.220).

A meta-analysis of two of the studies on PIBO cases that developed after an adenovirus infection showed no significant association between the duration of fever and the development of PIBO, using a logistic proportional hazard model [12,14]. However, the results of the meta-analysis from the other two studies, which used multivariate logistic regression analysis, showed that a longer duration of fever was significantly associated with the development of PIBO (OR, 1.128; 95% CI, 1.057–1.204) [9,13]

Four studies reported the association between the length of hospitalization and PIBO development [9,12,13,14]. Two studies showed a positive association between length of hospitalization and PIBO development after an adenovirus infection using a logistic proportional hazard model (HR, 1.070; 95% CI, 1.032–1.110) [12,14]. However, the other two studies [9,13] showed no significant associations (OR, 0.974; 95% CI, 0.939–1.001).

Two studies showed the association between an adenovirus infection and PIBO development using multivariate logistic regression analysis [9,15]. One study reported the association between adenovirus co-infection in children with MP pneumonia and PIBO development [9], and another study showed the association between an adenovirus infection and the development of PIBO [15]. The meta-analysis for these two studies showed a significant association between adenovirus infections and the development of PIBO (OR, 13.187; 95% CI, 5.450–31.911).

Three studies described the association between age (months) and the development of PIBO [9,13,15]. One study was excluded because it only reported prevalence in two age groups (<6 months of age and ≥6 months of age) [15]. The other two studies were included in the meta-analysis and showed no association between age and the development of PIBO.

## 4. Discussions

This systematic review and meta-analysis investigated the risk factors associated with the development of PIBO from seven studies that included HR or OR to minimize the heterogeneity arising from the diversity of the presented methods of the results in each study. Our meta-analysis showed that LDH levels, pleural effusion, hypoxemia, sex, and mechanical ventilation are risk factors for the development of PIBO among 9 variables that were mentioned in more than two studies. The significance differed for the length of hospitalization and duration of fever as risk factors for PIBO according to the statistical method used. In addition, the risk factors for developing PIBO differed according to the respiratory pathogen. The results of this study are helpful in the recognition of high-risk individuals that require follow-up to detect the development of PIBO with consideration of the respiratory pathogens and thereby improving the clinical outcomes of PIBO.

The pathogenesis of respiratory epithelium insults with related immune responses in response to respiratory infections differs according to the respiratory pathogens (Figure 3), which explains the differences in diagnostic and prognostic biomarkers and the risk factors of PIBO depending on the causative pathogens [16,17]. MP infection stimulates macrophages through Toll-like receptors and releases immunomodulatory and inflammatory cytokines and chemokines [18]. The exaggerated immune response is one of the key mechanisms of lung injuries in MP infections in that corticosteroids can be beneficial in treating severe or refractory MP infections [19,20], whereas corticosteroid treatment has no beneficial effect in respiratory viral infections [21]. The exaggerated immune response in MP infections is partially reflected in the elevated levels of LDH and pleural effusion [20,22,23], which were identified as risk factors for PIBO-associated MP infection in this meta-analysis. These findings suggest that an exaggerated immune response might be one of the pathophysiologies of PIBO after an MP infection. Changes in the characteristics of respiratory pathogens, such as an increasing trend of macrolide resistance of MP and refractory MP infections, which are becoming an important issue [24], might affect the development of PIBO with their characteristic pathophysiologic features. However, there have been no studies on these issues, partially due to the small number of PIBO patients and the lack of subsequent studies. Future studies on these issues are required to reveal the pathophysiologies of PIBO according to respiratory pathogens.

In association with the causative respiratory pathogens of the development of PIBO, adenovirus infections [15] and adenovirus co-infection in MP pneumonia [9] were associated with an increased risk of developing PIBO. The clinical manifestations of an adenovirus infection range diverse from asymptomatic to severe illness [25]. Some adenoviral illnesses cause severe lower respiratory tract infections even in healthy children and is linked with the development of PIBO in some cases [25]. The severe clinical course of respiratory infections, including adenovirus infections, are associated with mechanical ventilation and a longer duration of fever and hospitalization, which were identified as risk factors for the development of PIBO in this meta-analysis. In addition, the serotypes of the adenovirus might differently affect the development of PIBO, possibly through differences in the virulence according to the serotypes [26]. Only one study serotyped the adenoviruses in a part of the study population and showed that adenovirus 7 h was associated with the development of PIBO [11]. Although adenovirus types 5 and 21 are known to be associated with an increased risk of severe disease [25], there have been no studies on the association between these serotypes and the risk of developing PIBO. Future studies on these issues are needed to better understand the pathophysiologies of PIBO according to the respiratory pathogens and their serotypes.

In addition, host immunity and immunopathology against respiratory pathogens might affect the development of PIBO [27]. Among the identified potential risk factors of PIBO, age can be associated with host immunity. The meta-analysis of two studies identified the association between ages of less than six months and the development of PIBO, which revealed no significant associations [9,15]. Other analyses showed that age as a continuous variable was not a risk factor for the development of PIBO [9,13]. One recently published meta-analysis on the risk factors for PIBO showed that patients with PIBO were younger than the controls [28]. This meta-analysis included only studies with OR, HR, or RR, whereas the recently published meta-analysis included studies without considerations for the heterogeneity in the result values of included studies [28].

Regardless of respiratory pathogens as potential risk factors for PIBO, hypoxemia, mechanical ventilation, sex, and longer duration of fever and hospitalization were associated with increased risks for PIBO, when analyzed using logistic regression analysis. These findings suggest that severe and increased disease burden is related to the development of PIBO. Therefore, it is necessary to follow up with concerns on whether PIBO occurs after severe respiratory infections with increased disease burden resulting in respiratory tract infections.

This study has several limitations. First, the number of studies included in the meta-analysis was small. As clinical manifestations of PIBO are diverse, the period between onset and identification of PIBO by physicians varies from patient to patient, and the diagnosis of PIBO is often missed. The missed diagnoses lead to PIBO being reported as a rare disease [29], although the exact incidence and prevalence are unknown [1]. In addition, to reduce the heterogeneity of the results in the analyzed studies, only studies that reported the results using HR, OR, or RRs were included in this meta-analysis. As a result, only a few studies on the risk factors for PIBO were included in the meta-analysis of this study. The interpretation of the discrepancy of risk factors of PIBO according to statistical methods used in each study, such as the duration of hospitalization and fever, requires additional consideration of the differences in the results of the statistical analysis. Nevertheless, this is the first study to summarize the risk factors for the development of PIBO in children among studies that presented the risk factors using HRs, or ORs. Additionally, this is the first study to discuss the potential differences in the risk factors of PIBO according to respiratory pathogens.

In conclusion, we identified potential risk factors for the development of PIBO in children. Children with potential risk factors during respiratory tract infections are required follow-ups for the development of PIBO. These results can be useful in not missing a diagnosis of PIBO in children and therefore would help improve the clinical outcomes in children with PIBO.

## Figures and Tables

**Figure 1 pathogens-11-01268-f001:**
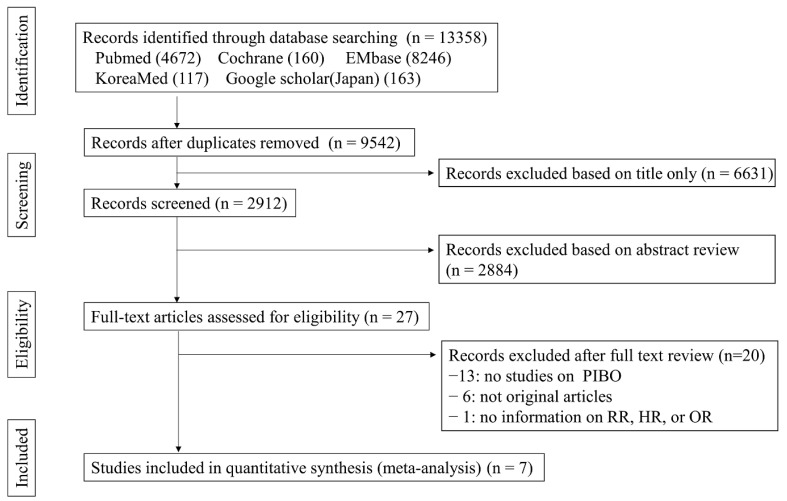
Preferred Reporting Items for Systematic Review and Meta-analyses (PRISMA) diagram for the literature search and study selection. HR, hazard ratio; PIBO, post-infectious bronchiolitis obliterans; OR, odds ratio; RR, relative risk.

**Figure 2 pathogens-11-01268-f002:**
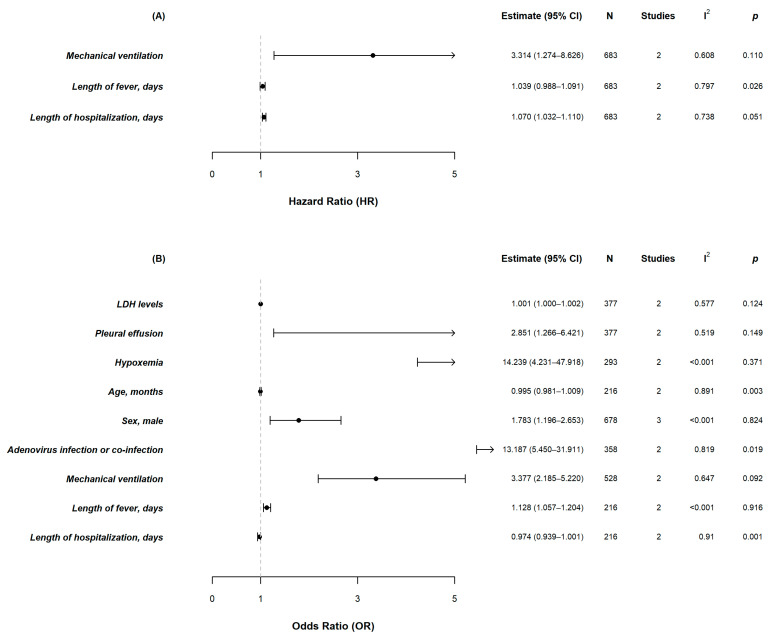
Forest plots of meta-analyses on the potential risk factors for the development of PIBO that were mentioned in two or more studies. (**A**) The meta-analysis was performed on three risk factors with HRs with 95% CIs; (**B**) The meta-analysis was performed on nine risk factors with ORs with 95% CIs.

**Figure 3 pathogens-11-01268-f003:**
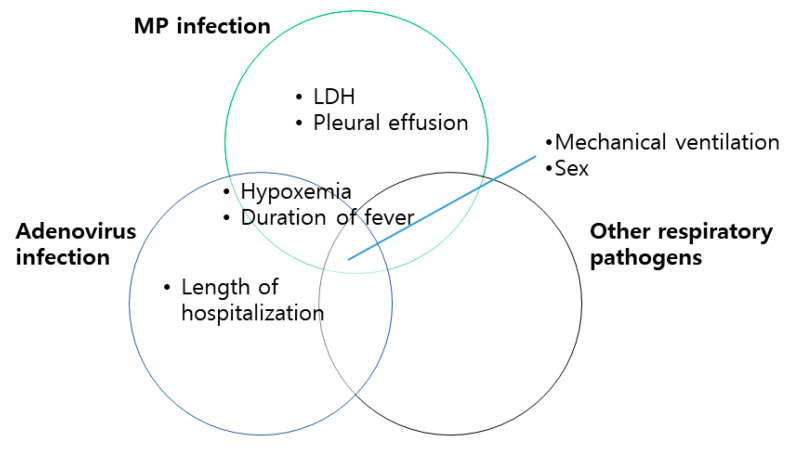
Risk factors of the development of PIBO according to respiratory pathogens.

**Table 1 pathogens-11-01268-t001:** Summaries of the potential risk factors for the development of PIBO, which were mentioned at least once in more than one study.

Risk Factors	Study (Year)	Variables	OR	Lower 95% CI	Upper 95% CI	*p* Value	Tau2	*I^2^*	Q Statistic	Degree of Freedom (Q)	*p* Value (Q)
** Respiratory virus co-infection **											
	Lee E et al. (2020) [9]	multivariate	4.069	1.224	13.523						
Summary statistics			4.069	1.224	13.527	0.022					
**Co-infections with RSV**											
	Murtagh P et al. (2009) [11]	univariate	4.100	1.500	10.600						
Summary statistics			4.100	1.500	11.207	0.006					
**LDH**											
	Lee E et al. (2020) [9]	multivariate	1.001	1.000	1.003						
	Huang K et al. (2012) [10]	multivariate	1.005	1.000	1.009						
Summary statistics			1.001	1.000	1.002	0.021	0.002	0.577	2.361	1.000	0.124
**Poor response to treatment**											
	Lee E et al. (2020) [9]	multivariate	41.760	2.792	624.543						
Summary statistics			41.760	2.792	624.574	0.007					
**Duration of moist rale**											
	Huang K et al. (2012) [10]	multivariate	1.203	1.066	1.358						
Summary statistics			1.203	1.066	1.358	0.003					
**Hypoxemia**											
	Huang K et al. (2012) [10]	multivariate	7.442	1.148	48.225						
	Yu X et al. (2021) [13]	univariate	22.846	4.633	112.666						
Summary statistics			14.239	4.231	47.918	<0.001	0.000	0.000	0.800	1.000	0.371
**Pleural effusion**											
	Lee E et al. (2020) [9]	multivariate	1.272	0.325	4.969						
	Huang K et al. (2012) [10]	multivariate	4.437	1.616	12.181						
Summary statistics			2.851	1.266	6.421	0.011	0.637	0.519	2.081	1.000	0.149
**Adenovirus infection or adenovirus co-infection**											
	Lee E et al. (2020) [9]	multivariate	5.607	1.801	17.454						
	Colom AJ et al. (2006) [15]	multivariate	49.000	12.000	199.000						
Summary statistics			13.187	5.450	31.911	<0.001	1.387	0.819	5.522	1.000	0.019
**Mechanical ventilation**											
	Colom AJ et al. (2006) [15]	multivariate	11.000	2.600	45.000						
	Murtagh P et al. (2009) [11]	univariate	3.000	1.900	4.600						
Summary statistics			3.377	2.185	5.220	<0.001	0.739	0.647	2.833	1.000	0.092
**>30 days of Hospitalization**											
	Murtagh P et al. (2009) [11]	multivariate	27.200	14.600	50.900						
	Summary statistics		27.200	14.600	50.673	<0.001					
**Length of hospitalization, days**											
	Lee E et al. (2020) [9]	multivariate	1.102	1.016	1.194						
	Yu X et al. (2021) [13]	univariate	0.944	0.906	0.985						
	Summary statistics		0.974	0.939	1.011	0.163	0.104	0.910	11.098	1.000	0.001
**Multifocal pneumonia**											
	Murtagh P et al. (2009) [11]	multivariate	26.600	5.300	132.000						
	Summary statistics		26.600	5.300	133.498	<0.001					
**Hypercapnia**											
	Murtagh P et al. (2009) [11]	multivariate	5.600	3.500	9.000						
	Summary statistics		5.600	3.500	8.960	<0.001					
**Persistent wheezing**											
	Yu X et al. (2021) [13]	multivariate	181.776	3.385	9761.543						
	Summary statistics		181.776	3.385	9760.737	0.011					
**Respiratory failure**											
	Yu X et al. (2021) [13]	multivariate	51.288	1.858	1415.441						
	Summary statistics		51.288	1.858	1415.661	0.020					
**Length of fever, days**											
	Lee E et al. (2020) [9]	multivariate	1.133	1.024	1.255						
	Yu X et al. (2021) [13]	univariate	1.125	1.033	1.226						
	Summary statistics		1.128	1.057	1.204	0.000	0.000	0.000	0.011	1.000	0.916
**Dyspnea**											
	Yu X et al. (2021) [13]	univariate	10.625	2.702	41.779						
	Summary statistics		10.625	2.702	41.779	0.001					
**Age (mo)**											
	Lee E et al. (2020) [9]	multivariate	0.990	0.976	1.003						
	Yu X et al. (2021) [13]	univariate	1.097	1.028	1.170						
	Summary statistics		0.995	0.981	1.009	0.452	0.069	0.891	9.148	1.000	0.003
**Sex, male**											
	Lee E et al. (2020) [9]	multivariate	1.570	0.569	4.329						
	Colom AJ et al. (2006) [15]	multivariate	1.250	0.385	5.000						
	Murtagh P et al. (2009) [11]	univariate	1.901	1.200	3.003						
	Summary statistics		1.783	1.196	2.653	0.005	0.000	0.000	0.387	2.000	0.824
**Adenovirus 7 h serotype**											
	Murtagh P et al. (2009) [11]	univariate	1.900	1.000	3.900						
	Summary statistics		1.900	1.000	3.610	0.050					
**Exposure to ETS at present**											
	Colom AJ et al. (2006) [15]	univariate	1.400	0.400	4.500						
	Summary statistics		1.400	0.400	4.900	0.599					
**Dyspnea**											
	Zhong L et al. (2020) [14]	multivariate	3.922	1.060	14.511						
	Summary statistics		3.922	1.060	14.511	0.041					
**Length of fever, days**											
	Wu PQ et al. (2016) [12]	multivariate	1.000	0.942	1.062						
	Zhong L et al. (2020) [14]	multivariate	1.129	1.033	1.234						
	Summary statistics		1.039	0.988	1.091	0.135	0.077	0.797	4.932	1.000	0.026
**Length of hospitalization, days**											
	Wu PQ et al. (2016) [12]	multivariate	1.044	0.999	1.091						
	Zhong L et al. (2020) [14]	univariate	1.129	1.058	1.205						
	Summary statistics		1.070	1.032	1.110	0.000	0.048	0.738	3.821	1.000	0.051
**Hypoxemia**											
	Wu PQ et al. (2016) [12]	multivariate	5.046	1.170	21.762						
	Summary statistics		5.046	1.170	21.762	0.030					
**Length of mechanical ventilation**											
	Zhong L et al. (2020) [14]	univariate	1.103	1.013	1.201						
	Summary statistics		1.103	1.013	1.201	0.024					
**Mechanical ventilation**											
	Wu PQ et al. (2016) [12]	multivariate	1.438	0.354	5.841						
	Zhong L et al. (2020) [14]	multivariate	6.861	1.854	25.390						
	Summary statistics		3.314	1.274	8.626	0.014	0.862	0.608	2.551	1.000	0.110

CI, confidence interval; ETS, environmental tobacco smoke; HR, hazard ratio; LDH, lactate dehydrogenase; OR, odds ratio; RSV, respiratory syncytial virus.

**Table 2 pathogens-11-01268-t002:** Characteristics of the included studies.

Study Author, Year	Country	Study Duration	Pathogens	Number		Age, Mean (SD)/Median (IQR)/Range		Risk Factors	Statistics
				Control	PIBO	Control	PIBO		
Lee E et al. (2020) [9]	South Korea	May 2019–February 2020	MP	132	18	6.1 (±3.9) y	4.8 (±2.6) y	Respiratory virus co-infection, duration between symptom onset and admission, LDH, poor response to treatment, adenovirus co-infection, length of fever	aOR
Huang K et al. (2012) [10]	China	January 2018–June 2020	MP	195	32	5 (3–6) y	5 (3–7) y	Duration of moist rale, LDH, hypoxemia, pleural effusion	aOR
Colom AJ et al. (2006) [15]	Argentina	1991–2002	NA	99	109	0–3 y	0–3 y	Adenovirus infection, mechanical ventilation	aOR
Murtagh P et al. (2009) [11]	Argentina	March 1998–May 2005	Adenovirus	203	117	11.2 (±10.6) y	10.5 (±8.8) y	>30 days of hospitalization, multifocal pneumonia, hypercapnia	aOR
Wu PQ et al. (2016) [12]	China	January 2011–December 2014	Adenovirus	530	14	23.5 (1–144) m	15.5 (6–72) m	Hypoxemia	HR
Yu X et al. (2021) [13]	China	October 2018–January 2020	Adenovirus	46	20	30.5 (17.0–50.8) m	16.5 (11.0–25.3) m	Persistent wheezing, respiratory failure	aOR
Zhong L et al. (2020) [14]	China	January 2015–February 2019	Adenovirus	105	34	20.5 (±14.6) m	15.1 (±7.2) m	Length of fever, dyspnea, invasive mechanical ventilation	HR

aOR, adjusted odds ratio; HR, Hazard ratio; IQR, interquartile range; LDH, lactate dehydrogenase; m, months; MP, *Mycoplasma pneumoniae*; NA, not applicable; PIBO, Post-infectious bronchiolitis obliterans; SD, standard deviation; y, years.

## Data Availability

Not applicable.

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
