# Peer review of "Risk Factors for the Development of Post-Infectious Bronchiolitis Obliterans in Children: A Systematic Review and Meta-Analysis"

_pathogens, 2022, doi:10.3390/pathogens11111268_

Round 1

Reviewer 1 Report

Dear Authors,

you have submitted a well-written systemic review and meta-analysis about the identification of risk factors for the development of post-infectious bronchiolitis obliterans in children. The number of studies included in meta-analysis was small but the statistical analysis was exclusively based on studies that reported estimates of odds ratio (OR), hazard ratio (HR) or relative risk (RR) with corresponding 95% confidence intervals (CIs) in order to reduce heterogeneity of the results. I have only some minor changes to denote:

1.      Line 58: please delete the second “include”.

2.      Line 197: please correct us such: “pathophysiologies”.

3.      Line 254: please delete the word “during” at the end of the sentence.

4.      Line 262: please correct us such: “bronchiolitis”.

5.      Line 275: please correct us such: “PIBO”.

Best Regards

Reviewer 2 Report

This systematic review and meta-analysis of risk factors for the development  of PIBO is a welcome addition the world literature on this subject. PIBO is an uncommon condition in children but causes significant morbidity and mortality in affected individuals. The authors are correct it is likely under and misdiagnosed, especially in countries with lower resources. The paper is well written and relevant.

Minor comments to authors

1.       Section 2.3  to 2.11  listing the individual variables reads quite difficult and appears to be redundant information that is presented in the tables. The authors should try reduced or summarise these sections that does not include repetitive descriptions of how many studies used OR vs HR etc.  Perhaps only a descriptive summary of pooled data analysis/findings?

2.       The authors rightly point about the pathophysiological differences between MPO and adenovirus aetiologies. This is interesting and worth highlighting as these pathogens appear to have different distributions in the work i.e MP is Asia vs adenovirus in South America.  It would be helpful to include a simple table or diagram showing the reader clearly risk factors associated by pathogen.

Reviewer 3 Report

Lee and colleagues undertook a very important task of finding risk-factors for the development of PIBO. Their work is interesting there are some issues however that need to be addressed;

1. What is the population studied? From the abstract I gathered that authors were investigating PIBO in children. If so it should be stated clearly in the title and in the material and methods section as well as in conclusions. If the studies also included adults this should be stated (with exact numbers) as well

2. Methods section should be before the results

3. Could the authors address why they did not include the studies with no OR or HR calculated? They should state how many studies were excluded solely for that reason. I would strongly recommend then to recalculate their findings (based on the additional findings).

4. Could the authors explain how their results are "beneficial for the diagnosis of PIBO and for the establishing management strategies in patients with risk factors" and how the "results highlight the importance of avoiding misdiagnosis"

5 Minor editorial issues - line 58 (including twice), line 254 - unfinished sentece

Round 2

Reviewer 3 Report

The article is ready for publication after English language editing.